# Inflammatory Myofibroblastic Tumor of the Upper Airways Harboring a New TRAF3-ALK Fusion Transcript

**DOI:** 10.3390/children8060505

**Published:** 2021-06-15

**Authors:** Valentina Di Ruscio, Angela Mastronuzzi, Ida Russo, Marianna Neri, Alessandra Stracuzzi, Isabella Giovannoni, Maria Luisa Tropiano, Maria Antonietta De Ioris, Giuseppe Maria Milano

**Affiliations:** 1Department of Hematology/Oncology, Cell and Gene Therapy, Bambino Gesù Children’s Hospital, IRCCS, Piazza Sant’Onofrio 4, 00165 Rome, Italy; valentina.diruscio@opbg.net (V.D.R.); angela.mastronuzzi@opbg.net (A.M.); ida.russo@opbg.net (I.R.); mantonietta.deioris@opbg.net (M.A.D.I.); 2Pediatric Unit Annunziata Hospital, Via Migliori 1, 87100 Cosenza, Italy; marianne76@libero.it; 3Pathology Unit, Bambino Gesù Children’s Hospital, IRCCS, Piazza Sant’Onofrio 4, 00165 Rome, Italy; alessandra.stracuzzi@opbg.net (A.S.); isabella.giovannoni@opbg.net (I.G.); 4Airway Surgery Unit, Pediatric Surgery Department, Bambino Gesù Children’s Hospital, IRCCS, Piazza Sant’Onofrio 4, 00165 Rome, Italy; marialuisa.tropiano@opbg.net

**Keywords:** inflammatory myofibroblastic tumor, children, ALK inhibitors

## Abstract

Inflammatory myofibroblastic tumor (IMT) is a rare disease that mainly involves the lung and the abdomen with an intermediate clinical course but a recurrence rate between 15–30%. Radical surgery represents the gold standard of treatment, while chemotherapy and radiotherapy are considered for unresectable lesions. The identification of *ALK* translocations in IMT opened the option for the use of target therapies. Indeed, the ALK inhibitors have changed the treatment approach for aggressive lesions, improving the prognosis. Intraluminal upper-way IMT is extremely rare and represents a medical challenge. We reported an endotracheal IMT case presenting a previously unknown *TRAF3-ALK* fusion transcript.

## 1. Introduction

An inflammatory myofibroblastic tumor (IMT) is a rare disease with a first peak before the age of 20 years and a second one between 50 and 60 years [1,2]. It was first described in 1973 as a primary lung tumor [3], and since then, both lung and multiple extrapulmonary manifestations have been reported [4,5,6].

The etiology remains unknown but probably is related to an abnormal inflammatory response due to an immunological trigger at different antigens.

A wide spectrum of clinical and biological behavior is described, ranging from benign proliferations to intermediate locally aggressive, intermediate rarely metastasizing, and malignant tumors [4].

The possibility of slow progression into a sarcoma has been reported [7], as well as metastatic spread [8]. The lung, soft tissues, and abdomen are the most involved primary sites. Surgery represents the stand-alone treatment for IMT, with a 91% 5-year disease-free survival [8]. Chemotherapy was considered for unresectable, multifocal, or metastatic disease with a response rate of 50–60%. Radiotherapy is usually reserved for a palliative approach, alone or in combination with chemotherapy [9,10]. Steroids or nonsteroid anti-inflammatory drugs have also been considered [7].

The *ALK* translocations are identified in IMT, representing an oncogenic trigger; the ALK inhibitors have changed the treatment approach for unresectable/metastatic and/or recurrent lesions, improving the prognosis and overall survival.

The endobronchial or endotracheal localization is extremely rare but with a challenging approach considering the efficacy of focal treatment.

We reported on an endotracheal IMT case with a *TRAF3-ALK* fusion transcript, reviewing published cases. To our knowledge, our case is the first with a *TRAF3-ALK* fusion transcript.

## 2. Case Report

A six-year-old girl was admitted to a general hospital with fever, persistent cough, and dyspnea. No episodes of recurrent respiratory infections were reported. Blood tests revealed only an increase of C-reactive protein 9.67 mg/dL (range 0–0.5); chest X-ray showed bilateral pneumonia. Differential diagnoses included inhalation of a foreign body, tuberculous pneumonia, and interstitial pneumonia.

Oxygen therapy, bronchodilating drugs plus steroids inhalers, and antibiotic therapy (with ceftriaxone (100 mg/kg endovenous) were started; rapidly, steroids were switched to endovenous administration, without any improvement; no antifungal therapy was taken into account. Extremely rapidly, the patient needed invasive respiratory support, without severe hypoxia. There were no practical difficulties in proceeding with orotracheal intubation and mechanical ventilation.

A computed tomography (CT) scan confirmed multiple pulmonary consolidations on both lobes. The Sars-Cov2 molecular nasopharyngeal test was negative. A solid-like parietal protrusion floating in the tracheal lumen (approximately 12 × 8 mm) was detected (Figure 1a). All microbiological tests were negative. Diagnostic workup included a fibroscopy with bronchiolar–alveolar washing and biopsy of the endotracheal mass.

The patient was referred to our hospital.

The pathology revealed an ulcerated mucosa with an underlying proliferation of bland spindle to stellate-shaped cells in a myxoid stroma associated with mild inflammatory infiltrate including lymphocytes, scattered plasma cells, and histiocytes. Immunohistochemical stains showed positivity for vimentin and smooth muscle actin (SMA), while ALK1, ALKp80, desmin, myogenin, cytokeratin CAM5.2, CD45, CD31, S100, EMA, and MUC4 were all negative. After obtaining informed consent for the genetic analyses, RNA was extracted from tumor formalin-fixed paraffin-embedded material, and NGS panel was performed using Archer^®^ Universal RNA Reagent Kit for Illumina^®^, Archer MBC adapters, and our custom-designed gene-specific primer (GSP) pool kit. NGS analysis allowed us to identify a novel in-frame *TRAF3-ALK* fusion, involving exon 10 of *TRAF3* and exon 20 of *ALK*. The breakpoints were at chr14:103369766 and chr2:29446208for *TRAF3* and *ALK*, respectively. The gene fusion was confirmed by RT-PCR and Sanger sequencing. Finally, a diagnosis of IMT was rendered. The surgery was postponed considering the high risk of bleeding, mutilation, and life-threatening complications. Crizotinib was started at 165 mg/m^2^/dose twice daily for 21 days/course, with a rapid improvement and weaning from mechanical respiratory support, confirmed at endotracheal fibroscopy, demonstrating a partial response (Figure 2).

The child was discharged from the pediatric intensive care unit (PICU). CRP values gradually reduced, with normalization after five days of therapy with crizotinib. After two-week treatment, a new CT scan showed a 70% reduction of the mass, achieving the best response after 4 weeks from crizotinib. No mild or severe treatment side effects were observed.

At the time of the last follow-up, after eight months of therapy, she is still on treatment. The patient is in good condition and achieved a complete response (Figure 1b).

## 3. Discussion

We described a rare case of pediatric endotracheal IMT with a not-previously reported *ALK* fusion transcript, successfully treated with ALK inhibitor.

Since the first report in 1989 [11], upper airways IMT-including trachea and main bronchus- have been reported in 16 patients. The median age at diagnosis was 9 years, with a male/female ratio of 1. All the patients have a histological diagnosis of IMT, but ALK status was reported in only five cases.

Concerning clinical symptoms, respiratory symptoms represent the major concern for emergency and pediatric wards; pneumonia and wheezing are the most frequent diagnosis, with an almost prompt response to treatment.

Treatment strategies are listed in Appendix A [11,12,13,14,15,16,17,18,19,20,21,22,23,24,25]. Surgery was the mainstay of therapy in the majority of patients, including either demolitive surgery [11,12,14,17,18,19,20,21,22,25] or more conservative approaches, with a subtotal resection [23], a sleeve resection [13,24], or a biopsy [16]. Surgery was followed by laser therapy in two cases [15,24], by a COX2-inhibitor therapy in the second one [16] and by a low-dose chemotherapy regimen [23] in the other.

Historically, surgery and chemotherapy were recommended, lacking a molecular characterization of IMT and, of course, ALK-inhibitors chance. Of note, surgical resection remains the standard of care, in order to control the local disease if performed without aggressive intent and without permanent damages to the patient. Even a relapse in the primitive site does not prejudge the possibility to obtain a prolonged remission if secondary disease local control is achieved [10]. In clinical practice, the surgical approach certainly is more feasible for accessible anatomical districts, while the upper airway tumors remain difficult to treat, and alternative therapeutic strategies need to be explored.

Chemotherapy is usually reserved as a second-line treatment among patients with advanced disease, administering chemotherapeutic agents typically effective for soft tissue tumors [26,27], with a response rate ranging from 64% to 54%, according to specific regimens.

The discovery of ALK mutations deeply changed the therapeutic approach to this disease [1]. In IMT, more than 10 different genes have been identified as *ALK* fusion partners, including *TPM3/4*, *RANBP2*, *TFG*, *CARS*, *ATIC LMNA*, *PRKAR1A*, *CLTC*, *FN1*, *SEC31A*, and *EML434* [28]. *ALK* status is known to correlate with survival [29]. Rarely, IMT can harbor mutations of *ROS1*, *PDGFRb*, *NTRK*, or *RET*, which needs further studies to correlate with clinical presentations and outcomes [30].

To our knowledge, the most common genetic alteration of *TRAF3* is deep deletion, followed by mutation and then amplification; truncation and fusion of *TRAF3* are less common but also detected in several different types of human cancers [31]. The fusion *TRAF3(exon10)-ALK(exon20)* identified in our patient has never been reported before.

During the last years, it appears clearly that the confirmation of the presence of an ALK gene fusion becomes mandatory in order to make the best therapeutic decision since ALK status is strongly correlated to prognosis [32].

The routine assessment of ALK expression by immunohistochemistry alone could not be enough due to a variable rate of ALK expression on the cell surface and the well-known difficulty of detection of several fusion partners with this methodic. Surely, the recent development and amelioration of molecular diagnosis such as next-generation sequencing (NGS) represent a significant goal for early diagnosis and appropriate treatment choice [33].

In our specific case, a prompt diagnosis with molecular characterization was achieved. Considering IMT diagnosis with a TRAF3-ALK fusion transcript, in an anatomical district that avoids a radical surgical resection, we decided to start crizotinib, with a rapid recovery and an impressive lesion reduction.

Crizotinib is a small molecule targeting multiples tyrosine kinases such as ALK, ROS, ROS1, MET, and interferes with ALK-pathway, blocking oncogenic proliferation [34].

It was approved firstly for advanced ALK-positive or ROS1-positive non-small cell lung cancer (NSCLC) [35]. In the last five years, the tyrosine kinases inhibitor’s results seem encouraging. The results of a phase I-II clinical trials were extremely promising: Mossè et al. detected an objective response rate for patients with ALCL of about 90%. In particular, among the 14 patients with an IMT diagnosis, an objective response was reported in 12 patients (7 CR and 5 PR achieving in 2 cases within 8 weeks and in 3 cases within 20 weeks of treatment), with an 86% response rate, suggesting its role as a front-line treatment for advanced or not-completely resectable tumors [34,35].

*ALK* fusions were also detected in neuroblastoma, rhabdomyosarcoma, anaplastic large-cell lymphoma, and IMT1 [36]; several trials investigating the safety and efficacy in these subsets were run.

In 2010, in a phase I/II clinical trial, Butrynski et al. reported a brilliant response to crizotinib of ALK-positive IMT [37].

Furthermore, a phase II pediatric clinical trial by the EORTC reported an objective response in 50% of patients, with mildly adverse events among 10% of patients (more frequently nausea, fatigue, blurred vision, and diarrhea, without any severe or life-threatening adverse events) [38].

Recently, further reports studied alternative ALK inhibitors, such as Alectinib and Ceritinib. Brivio et al. reported two cases of IMTs treated with a second-generation ALK inhibitor Ceritinib, with complete remission after two years of treatment, and in one case, with a quick relapse after its discontinuation but a prompt second remission after its restart [39]. Several other studies confirmed the promising results, achieving a partial or complete response with good compliance and no severe side effects [40,41,42,43,44].

Craig et al. performed a critical review of patients affected by ALK-positive IMT who underwent therapy with an ALK-inhibitor, including 29 patients. Of those, 12 experienced a complete response (41.3%), 14 a partial response (48.3%), and 3 (7%) a stable disease. Two (7%) had recurrence at the stop therapy; regardless, they achieved a second complete response after restarting the therapy [33].

In 2017, Koltsida et al. published, for the first time, a case of a 5-year-old male with a specific bronchial primary IMT. The patient underwent a bronchoscopic resection and suddenly start crizotinib for 3 months, achieving a partial response [45].

In conclusion, upper airway neoplastic lesions, as malformations, are a rare diagnosis but should be always excluded in absence of symptoms regression with the appropriate treatment; in these cases, an expert consultation using further diagnostic tools should be considered. Endoscopy or/and thorax CT scans are the major tools to be considered in patients without symptoms regression or improvement, often supported by mute history before the symptom presentation. A conservative approach should be addressed until the final histological diagnosis is completed.

An endotracheal localization is rare but should be considered in children with persistent respiratory symptoms after medical treatment and atypical findings at imaging evaluations. Among endotracheal tumors, IMT has a negligible occurrence but with a changed treatment strategy and outcome, due to the molecular characterization of the tumor over the last 10 years. The standard approach with primary demolitive surgery may be therefore delayed/omitted; considering the high risk of life-threatening complications related to anatomical localization, target therapy has opened a new area with an impressive response in targetable histology, allowing a conservative approach and the option of ALK inhibitors.

This report emphasizes the importance of considering different diagnoses in resistant/atypical pneumonia or wheezing. Moreover, a prompt and timing diagnosis may open new therapeutic options as first-line treatment. Target therapy achieves a quick control of respiratory distress without a demolitive surgical approach, ensuring a prolonged disease remission.

## Figures and Tables

**Figure 1 children-08-00505-f001:**
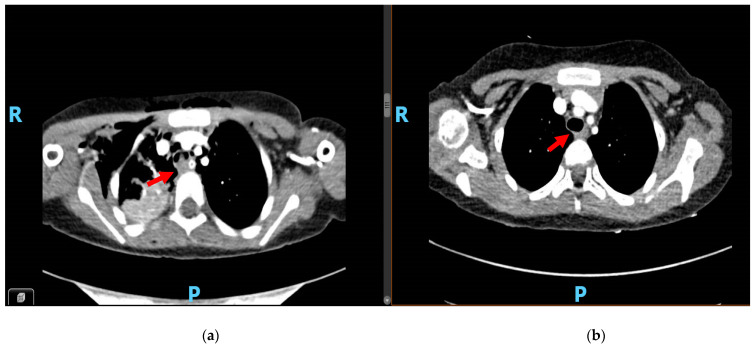
(**a**): CT scan at the diagnosis, showing the endotracheal lesion (red arrow). (**b**): CT scan shows the complete response achieving after eight months of therapy.

**Figure 2 children-08-00505-f002:**
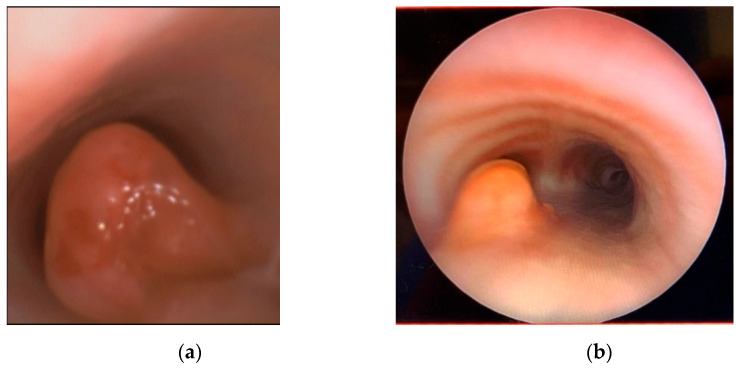
Shown in panels are (**a**) the fibroscopy at diagnosis and (**b**) after 2 weeks of crizotinib.

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
