# Peer review of "Inflammatory Myofibroblastic Tumor of the Upper Airways Harboring a New TRAF3-ALK Fusion Transcript"

_children, 2021, doi:10.3390/children8060505_

Round 1

Reviewer 1 Report

General comments
The current case is very important and interesting regarding the case of inflammatory myofibroblastic tumor of the upper airways harboring a new TRAF3-ALK fusion transcript.
There are several minor issues to be addressed for improving the content.

Specific comments
Major:
•    Couldn’t you recommend to have a one-stage excision strategy without a biopsy in this case?
•    How long do you think the follow-up period is appropriate?

Minor:
• Some references do not include the number of pages.
• The following documents could not be searched in PubMed.
45

Author Response

Thank you very much for your comments.

I try to answer to them briefly.

1) in this case, there were many differential diagnosis, such as an infectous pneumonia or an inhalation; therefore we started performing a biopsy in order to understand if it has an infectous or neoplastic origin and subsequently decide the correct treatment. If the neoplastic origin would have been the only one hypothesis suggested by radiological findings, we surely decided to perform a one-stage excision strategy.

2) I think that the correct follow-up should be among 10 years.

For the minor comments, I checked the bibliography for the number of pages. For reference 45, it is an abstract from the American Thoracic Society International Conference of 2017, I write the complete reference in the new version.

Reviewer 2 Report

Congratulations to the authors on reporting a non-surgical approach in the management of IMTs in the airway. While the novel approach is to be noted, there is not enough focus on the clinical presentation and work-up aspect and management of the patient. 

The article if improved with some more clinical data and following suggestions might be of better interest for the target audience of pediatric practitioners. 

Major Comments:

  1. What coverage of antibiotics were initiated? Oral vs parenteral. Were antifungals considered.
  2. What other differentials were considered?
  3. Was there inhaled bronchodilator therapy trialed prior to steroid administration?
  4. How hypoxic was the patient prior to "invasive respiratory support". Were there any complications to endotracheal intubation due to the tumor?
  5. ANy concerns with mechanical ventilation and/or anesthesia?
  6. Was CRP trended during the rest of the stay? If so what was the overall trend?

Minor comments:

  1. Line 56: " MA solid-like parietal protrusion". Is MA an abbreviation or a typo? Please cross-check.
  2. Line 187: Grammar could be rechecked: " This report stress over the importance" does not read well
  3. For figure 2: Arrows to tumor site might be helpful to readers with limited experience with CT images 

Author Response

Thank you very much for your congrats, and for your suggestions too.  I try to answer your comments:

1) Endovenous ceftriaxone (100 mg/kg for day) was started

2)  Differential diagnosis included inhalation, tuberculous pneumonia, and interstitial pneumonia.

3) Oxygen therapy, broncodilatators plus steroids inalhers and an antibiotic therapy (with ceftriaxone 100mg/kg endovenous) were started at the beginning; rapidly, steroids were switched into endovenous administration, without any improvement ; no antifungal therapy was taken into account.

4) Extremely rapidly, the patient needed an invasive respiratory support, without a severe hypoxia.

5) There was no practical difficulty in proceeding with orotracheal intubation, as well as for mechanical ventilation

6) CRP values gradually reduces, with a normalization after five days of therapy with Crizotinib. 

I checked for the minor comments that you suggested to us.

Round 2

Reviewer 2 Report

Thank you for addressing the suggestions. This gives the article more clinical credence in my humble opinion.